# Human Vδ2 T Cells and Their Versatility for Immunotherapeutic Approaches

**DOI:** 10.3390/cells11223572

**Published:** 2022-11-11

**Authors:** Marta Sanz, Brendan T. Mann, Alisha Chitrakar, Natalia Soriano-Sarabia

**Affiliations:** Department of Microbiology Immunology and Tropical Medicine, George Washington University, Washington, DC 20052, USA

**Keywords:** gammadelta cells, immunotherapy, allogeneic, expansion, cancer, HIV, infectious diseases

## Abstract

Gamma/delta (γδ) T cells are innate-like immune effectors that are a critical component linking innate and adaptive immune responses. They are recognized for their contribution to tumor surveillance and fight against infectious diseases. γδ T cells are excellent candidates for cellular immunotherapy due to their unique properties to recognize and destroy tumors or infected cells. They do not depend on the recognition of a single antigen but rather a broad-spectrum of diverse ligands through expression of various cytotoxic receptors. In this manuscript, we review major characteristics of the most abundant circulating γδ subpopulation, Vδ2 T cells, their immunotherapeutic potential, recent advances in expansion protocols, their preclinical and clinical applications for several infectious diseases and malignancies, and how additional modulation could enhance their therapeutic potential.

## 1. Background

γδ T cells are innate-like immune effectors with a critical role linking innate and adaptive immune responses owing to their unique properties [1,2]. Their defining feature is a T-cell receptor (TCR) comprised of variable γ and δ chains that recognizes non-peptide antigens in the absence of Major Histocompatibility Complex (MHC) molecules and elicit non-redundant functions when compared to conventional αβ T cells [1,2]. Human γδ T cells account for 0.5–10% of circulating T cells and are classified into two major subpopulations based on the δ chain usage, Vδ1 and Vδ2 T cells. The Vδ2 chain is present in the majority of circulating γδ T cells and is almost always paired with the Vγ9 chain (Vγ9Vδ2 T cells) [3,4].

γδ T cells are further characterized by membrane receptors and cytokine production capability that denotes a high degree of polyfunctionality as regulatory and effector cells [1]. γδ T cells shape the total immune response by secreting regulatory cytokines and interacting directly with other immune cell populations [5]. Each effector class plays a direct role in immune regulation and surveillance by modulating CD4^+^ T, CD8^+^ T, B, and dendritic cells (DCs) according to the challenge and the subset of γδ T cells involved in the response [6,7]. γδ T cells are also potent cytotoxic effectors, executing direct cytolytic activity mediated by the release of perforin, granzymes and Interferon- γ (IFN-γ) [1,8,9,10]. γδ T cell-mediated induction of apoptosis may occur through the expression of tumor necrosis factor receptor superfamily members TNF- related apoptosis-inducing ligand (TRAIL) or FasL, both of which have been demonstrated to be important for mediating the resolution of inflammation and clearing tumor cells [11,12]. In addition, expression of CD16 (FcγRIII) confers an additional mechanism for mediating antibody dependent cellular cytotoxicity (ADCC) [11,12,13,14].

Due to their unique biology, i.e., antigen recognition and effector functions, and our capacity to specifically induce their activation, γδ T cells are excellent candidates for cellular immunotherapy. Recognition and killing of tumors may not depend on the expression of a single antigen but diverse broad-spectrum ligands in cancer cells thanks to the expression of various cytotoxicity receptors [15]. Although the specific activating ligand for Vδ1 T cells remains elusive, an approved protocol for the clinical expansion of Vδ1 T cells developed by Silva-Santo’s group has provided new insights into the treatment of solid tumors and hematological malignancies [15,16]. A better understanding of Vδ2 T cell immunobiology has propelled most of the existing clinical trials. However, in the absence of a clinical protocol to expand Vδ2 cells ex vivo most clinical approaches have relied on the combined use of amino-bisphosphonates (N-BPs) and IL-2 or IL-15 as the preferred method to produce enough numbers (i.e., >90% Vδ2 T cells of total culture) for clinical application and further discussed below [8]. The potential to generate “super-killer” effector cells directed towards enhanced recognition and clearance is starting to be more widely investigated [17,18,19].

In this manuscript, we review major characteristics of Vδ2 T cells, their immunotherapeutic use for different adoptive cell therapy clinical applications in the context of malignancies and infectious diseases, discuss recent advances in expansion protocols and how additional modulation of Vδ2 T cell inherent properties could enhance their therapeutic potential.

## 2. Phosphoantigens for Ex Vivo Vδ2 T Cell Activation

The Vδ2 TCR specifically recognizes small “self” and “non-self” pyrophosphate antigens generally called phosphoantigens (p-Ags) [20,21,22]. “Non-self” p-Ags are microbial metabolites derived from the non-mevalonate or methyl-erythritol (MEP) pathway, including (E)-4-Hydroxy-3-methyl-but-2-enyl pyrophosphate (HMBPP) that triggers the most potent Vδ2 T cell activation [22,23]. Several laboratories have synthesized HMBPP with similar effects than the naturally derived HMBPP on Vδ2 T cells [24,25,26]. The use of HMBPP in combination with IL-2 leads to Vδ2 T cell proliferation and up-regulation of markers like CD16, CD25, or CD69 and secretion of pro-inflammatory cytokines [27,28]. Further combination of synthetic HMBPP in the presence of IL-21 enhances the antitumor cytolytic activity of Vδ2 T cells [23,29,30]. A second, less potent activator, isopentenyl pyrophosphate (IPP) is a prenyl pyrophosphate common to both the MEP and the Mevalonate (MVA) pathway that accumulates during periods of cellular stress [31,32]. 

The MVA pathway is an essential metabolic pathway common to all eukaryotic cells with critical roles in multiple cellular processes including synthesis of sterol derivatives such as cholesterol and non-sterol isoprenoids such as dolichol or heme-A [33]. Knowledge of the MVA pathway led to the development of several drugs such as statins and N-BPs that target different steps of the pathway [22,34,35,36,37,38] (Figure 1). Statins are widely used for the treatment of hypercholesterolemia, and they function by inhibiting the HMG-CoA reductase, which is the rate-limiting enzyme in the MVA pathway [39,40]. On the contrary, N-BPs are potent inhibitors of the synthesis of farnesyl pyrophosphate (FPP) and function by inhibiting the enzyme farnesyl pyrophosphate synthase [41]. N-BPs are the first line of treatment for osteoporosis and bone-related metastases [8,42]. These are small molecules with similar structure to prenyl pyrophosphates [43]. The use of N-BPs such as pamidronate (PAM), alendronate (ALN) or zoledronate (ZOL) has a dual effect, inducing accumulation of the natural antigen for Vδ2 T cells, IPP, and disrupting the production of FPP groups used for downstream processes such as protein prenylation. In the tumor setting, this inhibition provokes the disruption of important signal transduction pathways that regulate the proliferation, invasive properties, and pro-angiogenic activity of human tumor cells [44,45,46].

Although the specific mode of p-Ag presentation and recognition is not completely understood, the discovery of the involvement of transmembrane butyrophilin (BTN) molecules constituted an incredible breakthrough and exponentially increased our potential to exploit their intrinsic properties [47,48,49,50,51]. BTNs belong to the immunoglobulin superfamily of proteins with shared homology to B7 co-stimulatory molecules. Humans express seven distinct BTNs encoded by: BTN1A1, BTN2A1, BTN2A2, BTN2A3, BTN3A1, BTN3A2, and BTN3A3 as well as five additional butyrophilin-like (BTNL) proteins: BTNL2, BTNL3, BTNL8, BTNL9 and BTNL10 [52]. Each protein includes extracellular IgV and IgC domains and, with the exception of BTNL2 and BTN3A2, cytoplasmic B30.2 domains that are critical for intracellular signaling [49,53]. Specifically, the B30.2 domain on BTN3A1 binds to p-Ag leading to conformational changes that allow recognition by the Vγ9Vδ2 TCR and subsequent activation of Vδ2 cells [54,55]. Rigau M. et al., recently identified that, in addition to BTN3A1, BTN2A1 plays an important role as a ligand that binds to the Vγ9 TCR γ chain [48]. It is believed that once p-Ag binds to BTN3A1 through its intracellular domain, the BTN2A1–BTN3A1 complex engages the γδ TCR via two distinct binding sites: BTN2A1 binding to Vγ9 region, whereas another ligand (possibly BTN3A1) binding to the Vδ2 chain [48]. The use of monoclonal antibodies against BTN3 (CD277) can directly stimulate Vδ2 T cell responses and can be used to expand Vδ2 T cells both in vitro and in vivo [56,57]. The anti-BTN3 clone 20.1 specifically binds to an epitope within the extracellular IgV domain which generates or possibly stabilizes an “activating” conformation of dimeric BTN3A1. Additional agonistic antibodies such as ICT101 have been shown to promote Vδ2 T cell recognition of target cells, including through p-Ag independent mechanisms that require interaction between BTN3A and BTN2A isoforms [47,58]. The MHC-unrestricted nature of antigen recognition represents an advantage to develop allogeneic immunotherapies using Vδ2 T cells. However, there are also challenges associated with such approaches including culture conditions and interindividual variability. 

A previous study from our group showed that optimal in vitro expansion of Vδ2 T cells was achieved in the presence of MHC-II+ cells, and more specifically in cocultures with DCs [59]. A more recent study showed that depletion of αβ TCR+ cells during a two-step expansion protocol that included coculture with a genetically modified K562 cell line, induced an average fold expansion of around 230,000-fold [60]. 

An additional challenge for developing an adoptive cell therapy, includes interindividual Vδ2 subpopulation heterogeneity that may possibly require stratifying donors in terms of biological sex, age, ethnicity or even level of exercise to develop an optimal protocol that will provide sufficient numbers of cytotoxic Vδ2 T cells [61,62,63,64]. Several studies have investigated the variability of in vitro expansion rates among healthy individuals. Ryan PL. et al., elegantly showed a heterogeneous composition of proliferative versus cytotoxic Vδ2 functional subpopulations that was variable among healthy individuals [6]. In a follow up study, our group corroborated these findings in people living with HIV [64]. Interestingly, donor variability may be influenced by donor lifestyle, where exercise prior to PBMCs isolation has been shown to increase ex vivo expansion of γδ T cells [63]. In addition, the success of an expansion cannot be predicted based upon initial γδ T cell percentages [65], highlighting the complexity of finding optimal donors for expansion.

## 3. Clinical Applications of Vδ2 T Cell Immunotherapy

### 3.1. Immunotherapy for Infectious Diseases

Compared to oncology trials, discussed below, there is a dearth of clinical data for the therapeutic use of Vδ2 T cells to treat infectious diseases. To date, only a single pilot study has been published in which people living with HIV who had not yet started antiretroviral therapy (ART) received a short-course of ZOL and IL-2 [66]. The authors observed partial restoration in Vδ2 T cell functionality, including potential adjuvant effects boosting dendritic cell maturation and HIV-specific CD8 T cell responses. Data taken from preclinical animal models suggests that Vδ2 T cell immunotherapy may be unable to control early HIV replication despite possessing potent anti-HIV capabilities [67,68,69]. In contrast to acute infection, treating chronic SIV-infected Chinese rhesus macaques with HMBPP and IL-2 resulted in robust Vδ2 T cell expansion and pro-inflammatory cytokine production, which boosted both anti-HIV cellular and humoral responses [70]. Finally, our group has provided a proof-of-concept that Vδ2 T cells may be incorporated into current strategies for HIV cure within virally suppressed people living with HIV [71,72]. One approach utilized in HIV cure strategies encompasses reactivation of latent virus using small compounds generally called latency reversing agents and boosting immune responses [73,74,75,76,77,78]. We demonstrated that autologous expanded Vδ2 T cells mediated killing of latently HIV-infected CD4+ T cell reservoirs upon reactivation. Additional work from our lab showed that the frequency of cytotoxic Vδ2 + CD16+ cells inversely correlated with the recovery of replication-competent HIV from CD4+ T cell reservoirs [72]. This work corroborated the identification of proliferative versus cytotoxic Vδ2 T cell profiles in uninfected donors and people living with HIV [6,64]. Vδ2 T cell immunotherapeutic potential to cure HIV as well as other infectious diseases remains an intriguing avenue of research and has been previously reviewed by our group and others [76,79]. 

In several preclinical studies, administration of N-BPs or adoptive transfer of ex vivo expanded Vδ2 T cells demonstrated a therapeutic effect against both influenza and flavivirus infections (e.g., West Nile Virus and Hepatitis C Virus) [80,81,82,83]. Similar strategies were found to mediate the control of *Mycobacterium tuberculosis* infection in non-human primates [84,85]. Targeting Vδ2 T cell responses through vaccination is another possibility that has shown potential. Although historically neglected in vaccine research, the effect of the *Mycobacterium bovis* derived Bacillus Calmette-Guérin (BCG) vaccine on Vδ2 T cells has been well documented. BCG vaccination induces Vδ2 T cell expansion which is accompanied by increased IFN-γ production and changes in effector memory phenotypes that may confer protection to future infection [86,87,88,89]. Despite the lack of clinical validation in humans, these findings provide further evidence of the therapeutic potential of Vδ2 T cells against a wide breadth of pathogenic infections. 

### 3.2. Immunotherapy for Cancer

The critical importance of γδ T cell responses in immunosurveillance of both malignancies and pathogenic infection has been well established [90,91,92]. The ability of γδ T cells to exert potent cytotoxic functions in recognition of both self and foreign antigens, as well as stress ligands underscores their potential as a highly dynamic immunotherapeutic approach. Antitumor functions of γδ T cells have been extensively reviewed elsewhere and are only briefly mentioned here [1,8,93,94,95,96]. In addition to the TCR, γδ T cells recognize tumor cells through a variety of innate-like receptors involved in the recognition of stress molecules. Interestingly, receptor utilization for tumor killing is dependent on the specific tumor and microenvironment [97]. Stress-induced molecules upregulated in transformed cells including MHC class I polypeptide-related sequence A (MICA) or MICB bind to natural killer group 2 member D (NKG2D) [98]. Additional NK receptors like DNAM-1, natural cytotoxicity-triggering receptors NKp30 or NKp44 and TNF receptors are also involved in tumor recognition [99]. Expression of CD16 also enables γδ T cells to exert antibody-dependent cellular cytotoxicity upon engagement of membrane-surface antigens on the target cell bound to specific antibodies [100,101]. Upon ligation, γδ T cells produce cytotoxic molecules such as granzymes, perforins, and they can exert their killing functions through engagement of TNF-related apoptosis-inducing ligand (TRAIL) and Fas ligand (FasL) [102]. In addition to the direct cytotoxicity, γδ T cells provide necessary signals to induce responses from other cytotoxic effectors including production of cytokines that enables dendritic cell maturation as well as enhancement of CD8 T cell or NK cell killing [93,103,104]. 

The earliest clinical application of γδ T cells began with attempts to directly expand autologous Vδ2 T cells in vivo for individuals with late-stage malignancies that had exhausted previous lines of therapy. This came after observations that short course PAM and IL-2 could not only activate, but also boost Vδ2 T cell antitumor functions [36,105]. Several strategies have been utilized to harness Vδ2 T cells as an immunotherapy, differing on the source of donor cells, activating stimuli, and conditions for activation. Most of the early studies attempted Vδ2 T cell activation by directly administering N-BPs (e.g., PAM or ZOL) or synthetic P-Ag (e.g., BrHPP or 2M3B1PP) with low dose IL-2 to patients being treated for either solid tumors or hematological malignancies [106,107,108,109,110,111,112,113]. The use of synthetic P-Ags has the advantage of directly activating Vδ2 T cells rather than indirectly through inducing the accumulation of IPP upon treatment with N-BPs. In addition, synthetic P-Ags are more potent and require nanomolar concentrations, although its use has not been pursued probably due to the decreased number of Vδ2 T cells after prolonged use [113,114]. On the contrary, several studies have demonstrated the antitumor capacity of N-BPs by not only activating Vδ2 T cells, but also promoting apoptosis of sensitized cells [8,115]. In vivo treatment was generally well tolerated with only several observations of dose-limiting toxicities or high-grade adverse events possibly due to IL-2 toxicity. Despite robust expansion of peripheral Vδ2 T cells in many participants, anti-tumor activity was insufficient to significantly improve outcomes. Notably, repeated exposure to N-BPs or synthetic P-Ag throughout the course of treatment led to an increasingly exhausted phenotype of Vδ2 T cells, which may have limited any clinical benefit [66,107,108]. 

In an effort to improve efficacy and mitigate the aforementioned challenges observed in previous studies, researchers began manipulating autologous Vδ2 T cells ex vivo followed by reinfusion of expanded cells into the individual undergoing treatment, and studies are summarized in Table 1 [112,114,116,117,118,119,120,121,122,123,124,125,126,127,128,129]. In addition, a list of ongoing clinical trials targeting hematological malignancies have been recently reviewed [130]. The advantages to this approach include the ability to characterize the phenotype and anti-tumor potency of expanded Vδ2 T cells prior to administration to the participant. The phenotype of the expanded cell product has been characterized in several studies. The majority of ex vivo expanded γδ T-cells were CD27- CD45RA- cells [113,114,118,123,124,131] and had high NKG2D expression on their surface [123,124,125,131]. In several studies, expanded cells did not express PD-1 or CTLA-4, but did have detectable expression of TIM-3 and LAG-3 [123,125].

In addition, adoptively transferred cells could be radiolabeled to assess their ability to traffic to and penetrate solid tumors. Using indium [111] oxine, Nicol AJ et al., demonstrated that ex vivo expanded Vδ2 T cells are capable of penetrating metastatic sites within a participant with melanoma and another with lung adenocarcinoma [121]. Interestingly, the administered Vδ2 T cells displayed a similar pattern of tissue trafficking in each participant that received In-111 labeled cells. This was noted by rapid migration to the lungs during the first hour, which began to wane concurrently with their increased appearance in the liver and spleen by the fourth hour; a behavior that mimics adoptively transferred tumor-specific cytotoxic αβ T cells [132]. Similar to direct in vivo expansion, this strategy proved to be feasible and safe. Unfortunately, clinical outcomes varied between individual participants based on cancer type, disease stage, and previous treatment regimen. These factors may inherently limit the therapeutic use of autologous Vδ2 T cells within this context. Further studies are required to determine if there is a cancer-specific mechanism that impairs Vδ2 T cell proliferative and cytotoxic potential as well as individual effects of concurrent or previous therapies (e.g., chemotherapy, radiation, surgical resection, or additional immunotherapies).

In order to overcome the inherent limitations of autologous cells, we may exploit the MHC-independent mechanisms by which γδ T cells recognize and kill target cells. In the context of transplanting donor derived cells into a recipient (e.g., hematopoietic stem cell transplant), the genetic compatibility of the pair is paramount due to the risk of developing life-threatening Graft v. Host disease [133]. Studies on haploidentical HSCT revealed a positive association between allogeneic donor γδ T cells and disease-free survival in patients with late-stage hematological malignancies. This paved the way for separate trials investigating the direct infusion of ex vivo expanded allogeneic Vδ2 T cells [126,127,129,134,135]. These early stage trials published by Alnaggar. et al. and Xu et al. were pivotal in demonstrating the clinical proof of concept for an allogeneic Vδ2 T cell therapy using a solid tumor setting. Most notably, efficacy was demonstrated in cases by extended overall survival in the absence of severe adverse events for patients with lung or liver cancer. Positive changes in autologous αβ T cell and NK cell phenotypes were also observed, suggesting the infused cells may boost existing anti-tumor responses. Several of these observations were validated in more recent studies that explored allogeneic Vδ2 T cells to treat brain, pancreatic, or hepatic cancers [125,128,136]. Safety was consistent across each study, but efficacy remained mixed possibly due to factors such as previous or cotreatment, tumor burden, and systemic v. localized infusion. Despite these shortcomings, the robust responses observed in some individuals provides hope and justification for further investigation into allogeneic Vδ2 T cell-based immunotherapies for the treatment of cancer.

### 3.3. Lessons Learned from the Clinic

Regardless of the type of Vδ2 T cell therapy implementation, a common theme of variable clinical responses has been observed between previous trials. One of the more convincing arguments is based on evidence that suggests there are inherent interpersonal differences within Vδ2 T cell populations that directly impact their therapeutic potential. The Vδ2 T cell compartment is thought to be shaped by interaction with endogenous P-Ag during gestation as well as post-natal exposure to commensal bacteria [137,138,139]. Despite the prevalence of public CDR3γ9 clonotypes within Vδ2 T cell populations in adults, the influence of early life events on selecting clone lineages or skewing effector differentiation cannot be discounted. When factoring in the effects of age, gender, and previous P-Ag mediated activation, work by Ryan et al., and further work by our group, demonstrated that humans possess distinct subpopulations of Vδ2 T cells that differ in their cytotoxic and proliferative capabilities [6,64]. The variability of Vδ2 T cell responses within the clinic may be related to the polyclonal or homogeneous expansion of proliferative or cytotoxic subpopulations of Vδ2 T cells with mixed potency against tumor and infected cells [6]. 

The identification of an optimal donor profile or biomarker of highly responsive Vδ2 T cells may also lead to improved therapies. Examples of such potential biomarkers include CD16 expression associated with anti-HIV activity as demonstrated by our group [64,72] as well as increased NKG2A expression linked to hyperresponsive anti-tumor function as recently shown by Cazzetta et al. [19]. Additionally, overcoming donor heterogeneity may require optimizing the stimuli and culture conditions used within Vδ2 T cell expansion protocols, since currently we lack an accepted clinical protocol to specifically expand Vδ2 T cells. The different methods for Vδ2 T cell expansion are summarized in Table 2. Overcoming the variability of clinical responses is a critical step to make this approach widely available and a reality for treating a variety of diseases regardless of the origin. 

### 3.4. Enhancing Vδ2 T Cell Cytotoxic Effector Functions

The development of a clinical protocol may comprise strategic utilization of several approaches to produce adequate numbers of highly cytotoxic Vδ2 T effector cells. An increasing number of studies focused on γδ T cell immunotherapy has provided further insight into developing expansion protocols where specific interventions may be used to modulate Vδ2 T cell proliferation, phenotype, and function. To provide the proper context for interpreting these studies, it is important to understand which receptors mediate γδ T cell to recognition of target cells, their downstream effects on γδ T cell responses, and whether modulation of the expression of such receptors during expansion is possible.

γδ T cells recognize malignant cells through the engagement of innate activating receptors like NKG2D, and natural cytotoxicity receptors (NCRs) [1,155,156,157,158]. Therefore, inducing their expression may enhance recognition and possibly killing. NKG2D acts as a costimulatory signal upon recognition of stress markers, MHC class I chain related protein A and B (MICA, MICB) and UL16 binding proteins (ULBPs). Activation through this pathway provokes the secretion of proinflammatory cytokines TNF-α and IFN-γ as well as direct cytolytic activity mediated by the release of perforins and granzymes [159,160,161]. 

Several strategies could be used to upregulate NKG2D ligands on the surface of cancer cells. Temozolomide, used to treat brain tumors such as glioblastoma or Gemcitabine for bladder and breast cancer, increases the expression of NKG2D ligands making tumor cells more susceptible to recognition and lysis by γδ T cells [162,163]. Treatment with the histone deacetylase inhibitor, Valproic acid (VPA), significantly enhanced the expression of the NKG2D ligands MICA, MICB and ULBP-2, in a pancreatic carcinoma cell line (Panc89) and prostate carcinoma cell line (PC-3) [164]. Given the importance of NKG2D/ligands in cancer recognition and cytotoxicity, an expansion protocol that increases NKG2D expression on Vδ2 cells may be desirable. Dr. Kabelitz’s group demonstrated a differential capacity of HDACi to induce NKG2D expression on Vδ2 T cells, which could be an additional intervention for developing an optimized clinical expansion protocol [164].

Vδ2 cells can also express NKG2A which induces inhibitory signals through engagement with HLA-E/CD94 [165,166]. Interestingly, a recent study by Cazzetta, V. et al. demonstrated that expression of NKG2A identified a highly cytotoxic Vδ2 T cell subset against a variety of cancer cells from different origins. Furthermore, Vδ2 T cells expressing high levels of NKG2A supplied with remarkably higher cytokine production and cytotoxic potential compared to Vδ2 T cells that did not express NKG2A [19]. Since HLA-E is expressed virtually on all human tissues and allows for self-recognition, increased expression of HLA-E in several tumors constitute a mechanism of tumor escape that evades immune effector responses [167,168,169,170]. Therefore, extra attention should be given to changes in NKG2A expression during Vδ2 cell expansion and any attempts to select for this population will require additional measures to ensure the inhibitory nature of this receptor does not limit clinical efficacy. 

γδ T cell activity is also regulated by immune checkpoint receptors, such as program death-1 (PD-1), cytotoxic T-lymphocyte associated protein 4 (CTLA-4), lymphocyte-activation gene 3 (LAG-3), and T cell Immunoglobulin domain and mucin domain 3 (TIM-3) [171,172,173,174]. Blocking the inhibitory signaling mediated by these receptors with therapeutic monoclonal antibodies (mAbs) has proven beneficial for improving immune responses to refractory malignancies such as melanoma, non-small cell lung cancer, and renal cell carcinoma. This led to the development of FDA-approved mAbs targeting PD1/PD-L1, CTLA-4, and more recently LAG-3 [175,176]. The effects of immune checkpoint blockade (ICB) have primarily been studied within CD8 T cells, but several preliminary studies suggest a positive benefit may extend to γδ T cell-based immunotherapies. Although blockade of PD-1 does not appear to improve proliferation in vitro, it does result in increased IFN-γ production. Recent studies have shown that PD-1 checkpoint blockade enhanced anti-tumor immunity of adoptively transferred Vδ2 T cells in preclinical models and clinical trials [177,178,179]. Similarly, overexpression of TIM-3 on γδ T cells coincides with reduced expression of perforin and granzymeB and therefore decreased cytotoxicity against colon cancer cells [180]. Despite the lack of direct clinical evidence for ICB targeting γδ T cells, one study involving the CTLA-4 targeting mAb ipilimumab revealed a positive correlation between Vδ2 T cell frequencies and survival outcome for individuals who received the treatment [181]. These observations necessitate further investigation into harnessing ICB in order to improve γδ T cell immunotherapies. Activated Vδ2 T cells transiently upregulate the expression of PD-1 during ex vivo expansion with N-BPs [7,140,171,182]. TIM-3 and CTLA-4 expression has also been shown to be significantly up-regulated on Vδ2 T cells during expansion with ZOL which may contribute to a degree of cytotoxic dysfunction and increased susceptibility to apoptosis in a portion of expanded cells [183]. Importantly, effective blockade of PD-1 and TIM-3 during expansion may rescue γδ T cell cytotoxicity and effector function [183]. Despite the lack of ICB studies on γδ T cells, these collective preliminary findings suggest that blocking inhibitory receptors expressed on γδ T cells during expansion may provide an additional method of enhancing their cytotoxic functions.

### 3.5. Vδ2 T Cell Expansion: Combining the Cytokine Cocktail 

Vδ2 T cell activation and expansion requires a secondary signal from cytokines that affect Vδ2 T cell phenotype and functions (Figure 2).

Interleukins consist of a large group of proteins that can elicit distinct reactions in cells and tissues by binding to high-affinity receptors on the cell surface [184]. They play essential roles in the activation, differentiation, proliferation, maturation, migration, and adhesion of immune cells [184]. Below, we review cytokine use for the expansion of Vδ2 T cells, and the in vitro effect on Vδ2 T cell phenotype and function.

IL-2. IL-2 has been the most frequently used cytokine for the expansion of Vδ2 T cells. IL-2 is a growth factor essential for the proliferation of T cells and the generation of effector and memory cells [185,186]. N-BPs and p-Ags, in combination with IL-2 are strong stimulators of Vδ2 T cell activation and proliferation in ex vivo cultures of human PBMCs [28,36,140,141]. In an average of 10 to 14 days, Vδ2 T cells can expand between 40–70% depending on the N-BPs used [27,28,142,143,144,145]. Expansion protocols provided an initial exposure to N-BP or p-Ag in combination with IL-2 (100 and 1000 U/mL), replenished every 2–3 days, led to increased expression of cytotoxicity and activation markers like CD16, CD56, Nkp44, NKp30, and the production of IFN-γ, granzymeB, perforin, or TNF-α [22,28,141,142,146,150,151]. The anti-tumor functions of IL-2-expanded Vδ2 T cells against several tumor cancer cell lines have been investigated including human neuroblastoma, myeloma, glioblastoma, melanoma, and renal adenocarcinoma and Burkitt´s lymphoma [29,140,141,142,146,147,148]. 

IL-12. IL-12 has been studied extensively as a proinflammatory cytokine inducing production of IFN-γ by αβ T cells as well as γδ T cells. Expansion of Vδ2 T cells using a combination of IL-2, IL-12, and IL-18, induced over 80% IFN-γ producing Vδ2 T cells compared to a modest increase when using IL-2 combined with IL-15 [150,151]. The combination of IL-2, IL-12, and IL-18 also resulted in potent antitumor activity against different cell lines like Ewing sarcoma (A673), rhabdomyosarcoma (RH30), and neuroblastoma (SH-SY5Y) [150]. Upon IL-2, IL-12, and IL-18 stimulation, Vδ2 T cells also induced cell cycle arrest in various tumor cells like bladder carcinoma (T24), breast cancer (MCF7) and melanoma (WM115) [150]. 

IL-15. IL-15 plays a fundamental role in the stimulation of innate and adaptive cells, the reactivation of memory T cells, the induction of DC maturation, and the proliferation and activation of αβ T cells, γδ T cells, NK cells and B cells [187,188]. With an average of 6 to 14 days, studies have shown that Vδ2 T cells expanded between 30–60% depending on the p-Ag or NBPs used (IPP, BrHPP or ZOL) [18,27,140,152]. p-Ag combined with IL-15 significantly augmented T cell proliferation and intracellular cytokine expression, and together with IL-12, IFN-γ production was enhanced [152]. IL-15 also induced increased killing in ZOL or BrHPP-activated Vδ2 T cells against different tumor cell lines (Daudi and U299 cells) compared to IL-2 alone [140,189]. In addition, a higher killing was accompanied by an IL-15-mediated overexpression of cytotoxic markers and molecules like CD56, NKG2A, IFN-γ, TNF-α, and higher direct killing against several adherent tumor cell lines [18,180,189]. Similarly, IL-15 in combination with IL-2 boosted the proliferation as well as the in vitro anti-tumor activity of ZOL-expanded Vδ2 T cells [140]. 

IL-21. IL-21 is known to improve CD8^+^ T and NK cell cytotoxicity [29]. IL-21 combined with IL-2 promotes p-Ag-induced Vδ2 T cell proliferation [27,28,29]. In addition, IL-21 polarizes Vδ2 T cells toward a cytotoxic effector phenotype with an increase in granzymeA, granzymeB and perforin production levels [27,28,29]. The increase in cytotoxic activity is associated with enhanced tumor cell killing, although IL-21 neither synergizes nor interferes with IL-2 [29]. In acute myeloid leukemia patients, however, there was a reduction in the efficacy of IL-21 to increase proliferation and cytotoxicity of Vδ2 T, with a lower IL-21R and higher TIM-3 expression. Blocking TIM-3 could significantly improve the efficacy of IL-21 [190,191]. 

IL-18. IL-18 was discovered as an IFN-γ inducing factor and different studies demonstrated that it plays essential roles in host defense against infections and tumors, and in the pathogenesis of various inflammatory diseases through the upregulation of IFN-γ production [192,193,194]. IL-18 induced increased expansion of Vδ2 T cells stimulated by ZOL and IL-2, and expanded cells displayed an effector memory phenotype, with high levels of production of IFN-γ, TNF-α and strong cytotoxicity against tumor cells like mesothelioma cells [153,154]. As mentioned above, IL-18, in combination with IL-12, also increases proliferation, anti-tumor activity and IFN-γ production [150]. 

TGF-β. TGF-β is a pleiotropic cytokine that play roles in adult and embryonic growth and development, in inflammation and repair including angiogenesis and regulation of host resistance mechanisms [195]. TGF-β, combined with IL-2 or IL-15, increased the proliferation and cytotoxicity of BrHPP-activated Vδ2 T cells in vitro [196]. In addition, the expression of CD54, CD103, IFN-γ, IL-9 and granzyme-B were upregulated upon expansion. However, CD56 and integrin CD11a/CD18 were downregulated suggesting that transmigration into tissues may be impaired [18,197]. TGF- β expanded Vδ2 T cells were shown to have tumor cell killing against cell lines derived from pancreatic ductal adenocarcinoma (PDAC) like Colo357, MiaPaCa2, Panc1 and Panc89, as well as mammary gland adenocarcinoma cell line (MCF7) [18]. Vδ2 T cells activated with TGF-β have also shown to have better yield, viability, cytokine release and tumor cell killing compared to IL-2 activated controls [198]. However, TGF- β expanded Vδ2 T cells, displayed a more regulatory phenotype characterized by increased FOXP3 expression, and expression of PD-1, CTLA4 and CD80/CD86 were upregulated [7,8,149]. This suppressive function can be deemed counterproductive for use in immunotherapy.

Vitamin C. Vitamin C (L-ascorbic acid) is an important vitamin in many different biological processes, and it is a potent antioxidant and an essential cofactor in many enzymatic reactions [199,200]. Dr. Kabelitz’s group pioneered the investigation of the effect of Vitamin C and its more stable and less toxic derivative, L-ascorbic acid 2-phosphate on the proliferation and effector function effect of ZOL or synthetic p-Ags. Vitamin C and its analog reduced apoptosis and increased cytokine production during primary stimulation of a 14-day period of purified Vδ2 T cells [17]. The modulatory effect of vitamin C and its analog can be harnessed in order to optimize γδ T cell generation for cellular therapy [17].

## 4. Conclusions and Future Perspective

Overall, current developed protocols for clinical applications in oncology have achieved individual goals. However, a specific protocol strategically combining some approaches and combinations of cytokines may help achieve obtaining a universal protocol subjected to certain modifications based on the clinical use [16]. The clinical utilization of γδ T cells for infectious diseases, including HIV cure, remains understudied compared to the cancer field. Studies from our group and others paired with an examination of Vδ2 T cell basic biology warrants further investigation into their potential as an immunotherapy, beyond its utility against cancer [64,68,69,201].

The past two decades have provided examples of incredible progress in harnessing the power of the immune system in the form of life-saving cellular therapies. Despite cell therapy gaining acknowledgment as a new pillar of medicine, we are arguably still within the field’s infancy. Through the tireless efforts of pioneering researchers, we are well positioned to take the next step in developing γδ T cell-based immunotherapies through improved expansion protocols, possibly without requiring additional engineered genetic modifications. There are still a number of outstanding questions to answer both within the lab and in the clinic, but with a convergence of expertise from different fields, γδ T cells are poised to become the next tool in our armamentarium against disease.

## Figures and Tables

**Figure 1 cells-11-03572-f001:**
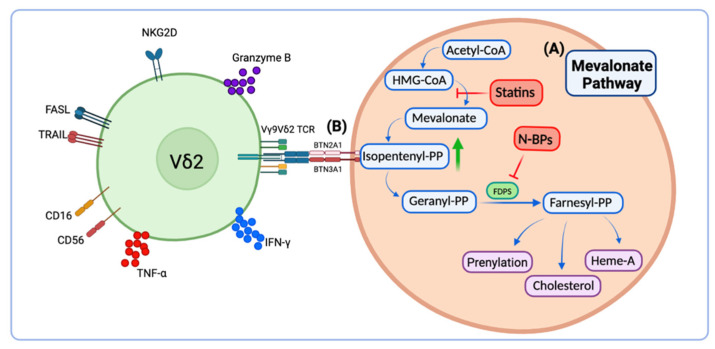
Mevalonate pathway and Vδ2 T cell-mediated activation. (**A**) Statins inhibit IPP production whereas N-BPs induce IPP accumulation that is (**B**) specifically recognized by the Vδ2 TCR in the context of butyrophilin molecules. (Created with Biorender, accessed on 10 December 2022).

**Figure 2 cells-11-03572-f002:**
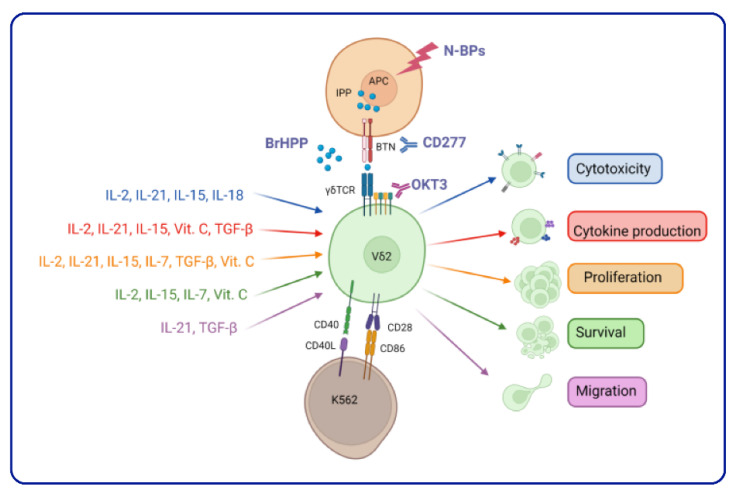
Modulation of Vδ2 T cell pleiotropic function. (Created with Biorender, accessed on 10 December 2022).

**Table 1 cells-11-03572-t001:** Summary of the clinical trials that used adoptive transfer of expanded Vδ2 T cells.

	Cancer Type	Approach	N	Expansion Method	Ref.
Autologous	Advanced RCC	Adoptive therapy	7	100 mM 2M3B1-PP + 100 U/mL IL-2, 14 days	[112]
	Advanced RCC	Adoptive therapy + low dose IL-2	10	3 mM BrHPP + 20–60 ng/mL IL-2, 14 days	[114]
	MM	Adoptive therapy	6	5 mM Zol + 1000 U/mL IL-2, 14 days	[116]
	RCC and multiple lung metastasis	Adoptive therapy + Zol + low dose IL-2	1	4 mg Zol + 1.4 million unit of IL-2	[117]
	NSCLC*	Autologous gd infusion	10	5 mM Zol + 1000 U/mL IL-2, 14 days	[118]
	Advanced RCC and lung metastasis	Adoptive therapy + Zol + low dose IL-2	41	100 mM 2M3B1-PP + 100 U/mL IL-2, 11 days	[119]
	NSCLC	Adoptive therapy	15	5 mM Zol + 1000 U/mL IL-2, 14 days	[120]
	Advanced solid tumors	Adoptive therapy + Zol	18	1 mM Zol + 700 U/mL IL-2	[121]
	Gastric	Adoptive therapy + Zol	7	5 mM Zol + 1000 U/mL IL-2, 14 days	[122]
	Pancreatic	Adoptive therapy + Gemcitabine	56	5 mM Zol + 1000 U/mL IL-2, 14 days	[123]
	NSCLC	Adoptive therapy	25	5 mM Zol + 1000 U/mL IL-2, 14 days	[124]
	Advanced pancreatic	Electroporation + adoptive therapy	62	50 mM Zol + 10 ng/mL IL-2	[125]
Allogeneic	Cancer type	Approach	N	Expansion method	Ref.
	Hematological malignancies	Haploidentical adoptive therapy	4	CD4 and CD8 depletion	[126]
	Cholangiocarcinoma, liver transplanted	Adoptive therapy	1	patent pending: not shown in paper	[127]
	Advanced HCC (N = 30)/ICC (N = 29)	Locoregional + adoptive therapy	59	50 mM Zol + 100 U/mL IL-2 + 100 U/mL IL-15 + 70 mM Vit C, 12–14 days	[128]
	Late-stage lung or liver	Adoptive therapy	18	50 mM Zol + 100 U/mL IL-2 + 100 U/mL IL-15 + 70 mM Vit C, 12–14 days	[129]

RCC: Renal cell cancer, NSCLC: non-small cell lung cancer, HCC: hepatocellular carcinoma, ICC: intrahepatic cholangiocarcinoma, MM: Multiple myeloma.

**Table 2 cells-11-03572-t002:** Summary of basic and clinical studies using phosphoantigens or bisphosphonates and different cytokines for Vδ2 T cell expansion.

CK	Concentration	pAg or BP	Additional Stimuli	Condition	Effect in γδ T cells	Target	Ref.
IL-2	50 U/mL	Zol (2.5 µM)	TGF-ß	PBMCs	IFN, GrzA, GrzB, Perforin, TNF, adhesion molecules	Pancreas and colon tumor cells	[18]
	100 U/mL	Synthetic HMBPP	-	PBMCs	IFN, TNF, IL5, IL-13	n/a	[28]
	20 ng/mL	HDMAPP	-	PBMCs	Proliferation, CD56, NKG2D/A, GrzA/B, Perforin	RCC and Burkitt’s lymphoma	[29]
	100 U/mL	IPP or ZOL	-	Isolated γδ	HLA-DR, CD69, CD56, CD16, IFN	Burkitt’s lymphoma and MM	[140]
	100 U/mL	ZOL (40µg/mL)	-	PBMCs	CD25, CD69, CD94, CD152, ICAM NKG2D, IFN, TNF	Human Neuroblastoma LAN 1	[141]
	10 U/mL	ZOL (1µM)	-	PBMCs	IFN, TNF	MM	[142]
	100 U/mL	IPP	aAPC, Anti-γδ TCR mAb	PBMCs/Isolated γδ	NKG2D, cytotoxicity	Neuroblastoma	[143]
	1000 U/mL	Zol (5 µM)	-	PBMCs	NKG2D, CD69, IFN	n/a	[144]
	100 U/mL	Zol (1µM)	IL-15	PBMCs	CD69, HLA-DR, CD80, CD86	n/a	[145]
	6.5 U/ML	IPP or TGF-B	Anti-γδ TCR mAb	Isolated γδ	GrzB, perforin, CD107a	Glioblastoma and melanoma	[146]
	400 U/mL	BrHPP (100 nM)	IL-33	PBMCs	CD16, CD28, NKG2A, NKG2D, Perforin, IFN, TNF, GrzB	Burkitt´s lymphoma	[147]
					CXCR3, CD28, CCR5, Trail		
	15.2 ng/mL	Zol (5 µM)	IL-15	Isolated γδ	NKG2D, NKG2A	Neuroblastoma cell lines	[148]
	50 U/mL	BrHPP (300 nM)	Vitamin C, TGF-ß	Isolated γδ	Proliferation	n/a	[149]
IL-12	10 ng/mL	IMMU510	IL2,IL-15,IL-18	Isolated γδ	IFN, TNF, GrzB, Perforin	sarcoma, rhabdomyosarcoma, neuroblastoma	[150]
	10 ng/mL		IL-18	PBMCs	IFN	n/a	[151]
IL-15	10 ng/mL	Zol (2.5 µM)	TGF-ß	PBMCs	IFN, GrzA, GrzB, Perforin, TNF, adhesion molecules	Pancreas and colon tumor cells	[18]
	10 ng/mL	HMBPP	-	PBMCs	Proliferation, TNF, IFN, CD16, CD94	n/a	[27]
	12.5 ng/mL	IPP or ZOL	-	Isolated γδ	Cytotoxicity, HLA-DR, CD69, CD56, CD16, IFN	Burkitt´s lymphoma and MM	[140]
	50 ng/mL	IPP or TGF-B	Anti-γδ TCR mAb	Isolated γδ	GrzB, perforin, CD107a	Glioblastoma and melanoma	[146]
	10 ng/mL	IPP (30µM)	IL-12	PBMCs	Proliferation, IFN,	n/a	[152]
IL-18	50 ng/mL	ZOL (1µM)	IL-2	PBMCs	IFN, TNF, cytotoxicity	Mesothelioma	[153]
	200 ng/mL	Zol (10 or 30 µM)	IL-1β, IL-2	PBMCs/Isolated γδ	Cytokine response	n/a	[154]
IL-21	10 ng/mL	HMBPP	-	PBMCs	Proliferation, CD16, CD94	n/a	[27]
	10 ng/mL	Synthetic HMBPP	-	PBMCs	CD25, CD27, CD69, NKG2D	n/a	[28]
	10 ng/mL	HDMAPP	-	PBMCs	Proliferation, CD56, NKG2D, GrzA, GrzB, Perforin	RCC and Burkitt´s lymphoma	[29]

RCC: Renal cell carcinoma, MM: Multiple myeloma, n/a: not applicable.

## Data Availability

Figures were created using Biorender, accessed on 10 December 2022.

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
