# Peer review of "Human Vδ2 T Cells and Their Versatility for Immunotherapeutic Approaches"

_cells, 2022, doi:10.3390/cells11223572_

Round 1
Reviewer 1 Report
Comprehensive review about Vd2 T cells.
Please highlight at one point and briefy in the review the therapeutic potential of Vd2 T cells is reviewed.
Author Response
Review 1:
In the last paragraph of the introduction, we mentioned that we review the therapeutic potential of Vd2 T cells (lines 58-62). We have also added this to the abstract.

Reviewer 2 Report
This is a timely and appropriate review of human Vδ2 cells for cancer immunotherapy. Especially, clinical applications for infection and cancer are well described. Although this review provides a lot of up-to-date information about human Vδ2 cells, it will help to add some more explanation for many unanswered questions.
1. A comprehensive description of memory and differentiation markers of Vδ2 cell activation and differentiation is required to follow their fate after clinical treatment.
2. The advantages and disadvantages of N-BP compared to non-self phosphoantigens need to be mentioned. N-BP leads to the accumulation of self-phosphoantigen IPP, which is less potent than non-self ones, and also inhibition of the protein prenylation pathway, which is not directly related to Vδ2 cells. Readers may want to know the potential benefits or harmful effects of non-self or synthetic phosphoantigens.
3. Detailed description of BTN family molecules regarding Vδ2 cells is necessary. Why does anti-BTN3 Ab stimulate Vδ2 cells? (lines 119-123) Does anti-BTN3 Ab change the BTN3 conformation for TCR ligation of Vδ2 cells? Some explanations are required.
4. NKG2A+ Vδ2 cells are reported to be a highly cytotoxic subset of Vδ2 cells. The authors mentioned that TGFβ upregulated the expression of NKG2A. But, NKG2A is an inhibitory NK cell receptor and its ligation is likely to inhibit the cytotoxic activity of NKG2A+ Vδ2 cells. The expansion of NKG2A+ Vδ2 cells may not be related to the upregulation of NKG2A. If the NKG2A expression marks the subsets of Vδ2 cells, the upregulation of the NKG2A expression will not be proportional to their cytotoxic potential. Furthermore, TGFβ may be immunosuppressive or pro-tumoral by activating Tregs or TH17 cells. The authors need to clarify this point. This point needs to be mentioned in the section on Vδ2 cell expansion.
Author Response
Review 2:
This is a timely and appropriate review of human Vδ2 cells for cancer immunotherapy. Especially, clinical applications for infection and cancer are well described. Although this review provides a lot of up-to-date information about human Vδ2 cells, it will help to add some more explanation for many unanswered questions.
- A comprehensive description of memory and differentiation markers of Vδ2 cell activation and differentiation is required to follow their fate after clinical treatment.
As suggested, we have now added a summary of the literature describing the phenotype of ex vivo expanded Vd2 T cells for adoptive transfer (Lines 242-246)
- The advantages and disadvantages of N-BP compared to non-self phosphoantigens need to be mentioned. N-BP leads to the accumulation of self-phosphoantigen IPP, which is less potent than non-self ones, and also inhibition of the protein prenylation pathway, which is not directly related to Vδ2 cells. Readers may want to know the potential benefits or harmful effects of non-self or synthetic phosphoantigens.
As suggested, we have included a brief comparison between N-BPs and P-Ags utilized in clinical trials (lines 223-229)
- Detailed description of BTN family molecules regarding Vδ2 cells is necessary. Why does anti-BTN3 Ab stimulate Vδ2 cells?Does anti-BTN3 Ab change the BTN3 conformation for TCR ligation of Vδ2 cells? Some explanations are required. As suggested, we have added more description of the BTN family and antigen recognition (lines 113-132)
- NKG2A+ Vδ2 cells are reported to be a highly cytotoxic subset of Vδ2 cells. The authors mentioned that TGFβ upregulated the expression of NKG2A. But, NKG2A is an inhibitory NK cell receptor and its ligation is likely to inhibit the cytotoxic activity of NKG2A+ Vδ2 cells. The expansion of NKG2A+ Vδ2 cells may not be related to the upregulation of NKG2A. If the NKG2A expression marks the subsets of Vδ2 cells, the upregulation of the NKG2A expression will not be proportional to their cytotoxic potential. Furthermore, TGFβ may be immunosuppressive or pro-tumoral by activating Tregs or TH17 cells. The authors need to clarify this point. This point needs to be mentioned in the section on Vδ2 cell expansion.
We apologize for the confusion. We have now clarified this point further in lines 382-392

Reviewer 3 Report
gd T cells are high potential candidates for clinical cellular immunotherapy owing to their unique properties of recognizing and destroying tumor cells of malignancies or pathogens of infectious diseases. Throughout the manuscript, Sanz et al. provided a comprehensive summary for current clinical applications of immunotherapy regarding Vd2 T cell, the most abundant circulating gd subpopulation. The authors summarized important cutting-edge findings, including Vd2 T cell biology, expansion protocols, and clinical applications of Vd2 T cell immunotherapy for both infectious diseases and cancers, as well as the caveats from the clinic. Overall, the manuscript is well-written and informative. The authors provided excellent insights and potential directions of research gaps for future advanced research to further enhance the therapeutic potential of Vd2 T cell involvement in immunotherapy.
It remains one section that needs to be further clarified:
1). In figure 1, the description of figure legend seems to be conflict with the context., eg. N-BPs induce IPP accumulation, which was not consistent with the description in “line 88-91”. Besides, the direction of red block-line ( ―l ) for both Statins and N-BPs seems not match the descriptions in the context. Please confirm and correct it if it is incorrect.
Author Response
It remains one section that needs to be further clarified:
1). In figure 1, the description of figure legend seems to be conflict with the context., eg. N-BPs induce IPP accumulation, which was not consistent with the description in “line 88-91”. Besides, the direction of red block-line ( ―l ) for both Statins and N-BPs seems not match the descriptions in the context. Please confirm and correct it if it is incorrect.
We thank the reviewer for catching this error. We have now corrected the figure accordingly.

Round 2
Reviewer 2 Report
Major issues were addressed properly.